

# Hyperspectral tree crown classification using the multiple instance adaptive cosine estimator

Sheng Zou[1], Paul Gader[2] and Alina Zare[1]

[1] Department of Electrical and Computer Engineering, University of Florida, Gainesville, FL, United States of America
[2] Department of Computer & Information Science & Engineering, University of Florida, Gainesville, FL, United States of America

## ABSTRACT

Tree species classification using hyperspectral imagery is a challenging task due to the high spectral similarity between species and large intra-species variability. This paper proposes a solution using the Multiple Instance Adaptive Cosine Estimator (MI-ACE) algorithm. MI-ACE estimates a discriminative target signature to differentiate between a pair of tree species while accounting for label uncertainty. Multi-class species classification is achieved by training a set of one-vs-one MI-ACE classifiers corresponding to the classification between each pair of tree species and a majority voting on the classification results from all classifiers. Additionally, the performance of MI-ACE does not rely on parameter settings that require tuning resulting in a method that is easy to use in application. Results presented are using training and testing data provided by a data analysis competition aimed at encouraging the development of methods for extracting ecological information through remote sensing obtained through participation in the competition. The experimental results using one-vs-one MI-ACE technique composed of a hierarchical classification, where a tree crown is first classified to one of the genus classes and one of the species classes. The species-level rank-1 classification accuracy is 86.4% and cross entropy is 0.9395 on the testing data, provided by the competition organizer, without the release of ground truth for testing data. Similarly, the same evaluation metrics are computed on the training data, where the rank-1 classification accuracy is 95.62% and the cross entropy is 0.2649. The results show that the presented approach can not only classify the majority species classes, but also classify the rare species classes.

# INTRODUCTION

Hyperspectral remote sensing is an active area of research with studies in the literature spanning classification (*Demir & Erturk, 2007*; *Li et al., 2014*), unmixing (*Bioucas-Dias et al., 2012*; *Zou & Zare, 2017*; *Zare & Ho, 2014*) and segmentation (*Tarabalka, Chanussot & Benediktsson, 2010*; *Li, Bioucas-Dias & Plaza, 2012*). This work focuses on the classification of tree crowns in hyperspectral remotely sensed imagery to tree species. Spectral signatures

Corresponding author
Alina Zare, azare@ufl.edu

of tree crowns across species often have high spectral similarity as well as significant intra-species variability (*Cochrane, 2000*), making tree crown classification from hyperspectral imagery a challenging task. In this work, a discriminative multiple instance hyperspectral target characterization method, the one-vs-one Multiple Instance Adaptive Cosine Estimator (MI-ACE) algorithm (*Zare, Jiao & Glenn, 2018*), is proposed for this problem.

In general, supervised machine learning algorithms can be divided into two steps: *training* and *testing*. During *training*, a set of training data and associated training desired output labels are used to learn a mapping from the input training data to the desired output labels. In our application, the training data are the pixels (i.e., the hyperspectral signatures corresponding to each spatial location in the image) and their correspoding training labels are their corresponding tree species (e.g., *Pinus palustris*, *Quercus laevis*). In the *testing* step, this mapping is then used to predict the species classes for unseen, unlabeled test data. However, precise pixel level training labels are often expensive or, even, infeasible to obtain (*Blum & Mitchell, 1998*). In the case of tree crown classification, pixel level ground truth labeling for tree crowns can be extremely challenging to collect. When looking at aerial hyperspectral imagery, given overlapping tree crowns and *mixed pixels* in which individual pixels contain responses from multiple neighboring tree species due to the image spatial resolution, manually labeling individual tree crowns is generally infeasible as the precise outline of each tree cannot be easily identified. *Marconi et al. (2018)* organized data science challenges for airborne remote sensing data. One of these challenges was to perform species classification of individual trees given airborne hyperspectral data. The challenge provided competitors (*Anderson, 2018*; *Sumsion et al., 2018*; *Dalponte, Frizzera & Gianelle, 2018*) with training and testing hyperspectral signatures extracted from individual tree crowns in the National Ecological Observatory Network (NEON) hyperspectral data collected at the Ordway-Swisher Biological Station in north-central Florida. These signatures were extracted from the imagery and labeled by the competition organizers by generating individual tree crown polygons using a tablet computer, GIS software, and an external GPS device in the field as described by *Marconi et al. (2018)*. The team loaded the aerial hyperspectral imagery onto tablet computers in the field and simultaneously visually assessed the scene in person and the aerial imagery to mark and digitize the outlines of individual tree crowns. This was a time consuming process that required some subjectivity in assessing the field and the overhead view. The difficulty of this process and the subjectivity needed may result in some individual pixels in tree crown polygons being mislabeled. It is assumed that there is at least one pixel that is labeled correctly in each individual tree crown (although we do not know which one(s)). This type of label imprecision is known as Multiple Instance style labels in the machine learning literature (*Maron & Lozano-Pérez, 1998*). The MI-ACE algorithm presented in this paper is designed to be robust to this sort of imprecise label without the need for any parameter tuning or any additional steps for outlier removal.

MI-ACE is a multiple instance learning (MIL) algorithm (*Maron & Lozano-Pérez, 1998*) where precise instance level labels are not necessary. Instead only a *bag* level label indicating the existence or absence of a target in a bag (or set) of instances is needed. In MIL, a bag is labeled as a *positive bag* containing a target if at least one data point in the bag corresponds

to target and a bag is labeled as a *negative bag* if none of the data in the bag correspond to the target. The MIL problem was first proposed and investigated by *Dietterich, Lathrop & Lozano-Pérez (1997)* where he proposed to use axis-parallel rectangles in the feature space to identify and define the target concept in MIL. Here, target concept refers to a representation of the target class in the feature space (e.g., in the case of hyperspectral target detection, the target concept can be a target spectral signature). Since this initial approach, several MIL methods have been published in the literature. These include the method by *Maron & Lozano-Pérez (1998)* who proposed a Diverse Density (DD) framework for solving the target concept estimation problem in MIL. Inspired by DD, *Zhang & Goldman (2002)* introduced an Expectation-Maximization (EM) optimization method to DD.

As opposed to previous methods which generally try to reconstruct a representative target concept, the MI-ACE algorithm estimates a *discriminative* target signature from data with this sort of bag-level MIL labels. The discriminative target signature highlights differences between the target and non-target classes in the data. This target signature can then be used within the ACE detector (*Kraut & Scharf, 1999*; *Kraut, Scharf & McWhorter, 2001*; *Basener, 2010*) to perform pixel-level target detection and classification. As compared with other hyperspectral target detection methods (*Key, 1993*; *Manolakis, Marden & Shaw, 2003*; *Nasrabadi, 2008*; *Matteoli, Diani & Theiler, 2014*; *Theiler & Foy, 2006*), MI-ACE does not need a target signature in advance. In addition, MI-ACE does not assume a pure pixel representations of the target in the training data, but instead is able to detect subpixel targets. More importantly, as previously stated, MI-ACE does not need precise pixel-level labels. Since MI-ACE needs only bag-level labels, the algorithm naturally addresses the tree crown classification problem outlined above. Each tree crown (and the associated set of hyperspectral signatures) are considered a bag and that bag is labeled as the corresponding target tree species. Since MI-ACE assumes multiple instance style labels, the algorithm does not assume that each pixel in every bag is representative of the associated tree species (but only assumes that there exists at least one representative signature in the tree crown) and, thus, addresses imprecision in labeling. These advantages of MI-ACE make it fit the species classification problem over other methods, since for species classification on hyperspectral images, (1) the target signature of each species is often difficult to obtain; (2) for some tree crowns, there are a few pixels or even only a few subpixels containing the target class; (3) labeling each pixel in hyperspectral image is expensive and difficult.

However, the current MI-ACE algorithm is limited in that it only applies to binary classification problems. Thus, in this paper, MI-ACE is extended to fit the multi-class classification problem with a proposed one-vs-one multi-classifier strategy. The one-vs-one strategy is commonly used for generalizing a binary-class classification problem to multi-class classification problems (*Milgram, Cheriet & Sabourin, 2006*; *Quinlan, 1986*; *Akhter, Heylen & Scheunders, 2015*; *Kwon & Nasrabadi, 2006*; *Nasrabadi, 2008*).

## METHODS

In this section, a brief review of MI-ACE is presented, then the proposed one-vs-one MI-ACE tree species classification approach is outlined.

## MI-ACE target characterization

MI-ACE (*Zare, Jiao & Glenn, 2018*) is a discriminative target characterization method that is based on ACE detector and multiple instance concept learning. In multiple instance learning, sets of data points (termed bags) are labeled as either positive or negative. Specifically, let $\mathbf{X} = [\mathbf{x}_1, \ldots, \mathbf{x}_N] \in \mathbb{R}^{d \times N}$ be training data where $d$ is the dimensionality of an instance, $\mathbf{x}_i$, and $N$ is the total number of training instances. The data are grouped into $K$ bags, $\mathbf{B} = \{\mathbf{B}_1, \ldots, \mathbf{B}_K\}$, with associated binary bag-level labels, $L = \{L_1, \ldots, L_K\}$ where $L_j \in \{0, 1\}$ and $\mathbf{x}_{ji} \in \mathbf{B}_j$ denotes the $i$th instance in bag $\mathbf{B}_j$. Positive bags (i.e., $\mathbf{B}_j$ with $L_j = 1$, denoted as $\mathbf{B}_j^+$) contain at least one instance composed of some target:

$$\text{if } L_j = 1, \exists \mathbf{x}_{ji} \in \mathbf{B}_j^+ \text{ s.t. } \mathbf{x}_{ji} \sim \mathcal{N}\left(\alpha_{it}\mathbf{s} + \boldsymbol{\mu}_b, \sigma_1^2 \Sigma_b\right), \alpha_{it} \neq 0 \tag{1}$$

where $\Sigma_b$ is the background covariance, $\mu_b$ is the mean of the background, $\mathbf{s}$ is the known target signature which is scaled by a target abundance, $a$, and $\sigma_1^2 = \frac{1}{d}(\mathbf{x} - a\mathbf{s})^T \Sigma_b^{-1}(\mathbf{x} - a\mathbf{s})$. However, the number of instances in a positive bag with a target component is unknown. If $\mathbf{B}_j$ is a negative bag (i.e., $L_j = 0$, denoted as $\mathbf{B}_j^-$), then this indicates that $\mathbf{B}_j^-$ does not contain any target:

$$\text{if } L_j = 0, \mathbf{x}_{ji} \sim \mathcal{N}\left(\boldsymbol{\mu}_b, \sigma_1^2 \Sigma_b\right) \forall \mathbf{x}_{ji} \in \mathbf{B}_j^- \tag{2}$$

Given this problem formulation, the goal of MI-ACE is to estimate the target signature, $\mathbf{s}$, that maximizes the corresponding adaptive cosine estimator (ACE) detection statistic for the target instances in each positive bag and minimize the detection statistic over all negative instances. This is accomplished by maximizing the following objective:

$$\arg\max_{\mathbf{s}} \frac{1}{N^+} \sum_{j:L_j=1} D_{ACE}(\mathbf{x}_j^*, \mathbf{s}) - \frac{1}{N^-} \sum_{j:L_j=0} \frac{1}{N_j^-} \sum_{\mathbf{x}_i \in B_j^-} D_{ACE}(\mathbf{x}_i, \mathbf{s}) \tag{3}$$

where $N^+$ and $N^-$ are the number of positive and negative bags, respectively, $N_j^-$ is the number of instances in the $j$th negative bag, and $\mathbf{x}_j^*$ is the selected instance from the positive bag $B_j^+$ that is mostly likely a target instance in the bag. The selected instance is identified as the point with the maximum detection statistic given a target signature, $\mathbf{s}$:

$$\mathbf{x}_j^* = \arg\max_{\mathbf{x}_i \in B_j^+} D_{ACE}(\mathbf{x}_i, \mathbf{s}) \tag{4}$$

Since the first term of the objective function relies only on the selected instance from each positive bag, the method is robust to outliers and incorrectly labeled samples. In other words, a target point will have a small spectral distance to the target signature and, thus, likely to be identified as the selected instance for a bag whereas outliers will be far from the target signature and disregarded. The $D_{ACE}$ is the ACE detection statistic,

$$D_{ACE}(\mathbf{x}, \mathbf{s}) = \left(\frac{\hat{\mathbf{s}}}{\|\hat{\mathbf{s}}\|}\right)^T \left(\frac{\hat{\mathbf{x}}}{\|\hat{\mathbf{x}}\|}\right) = \hat{\mathbf{s}}^T \hat{\mathbf{x}} \tag{5}$$

where $\hat{\mathbf{x}} = \mathbf{D}^{-\frac{1}{2}} \mathbf{U}^T (\mathbf{x} - \boldsymbol{\mu}_b)$ and $\hat{\mathbf{s}} = \mathbf{D}^{-\frac{1}{2}} \mathbf{U}^T \mathbf{s}$, $\mathbf{U}$ and $\mathbf{D}$ are the eigenvectors and eigenvalues of the background covariance matrix, respectively.
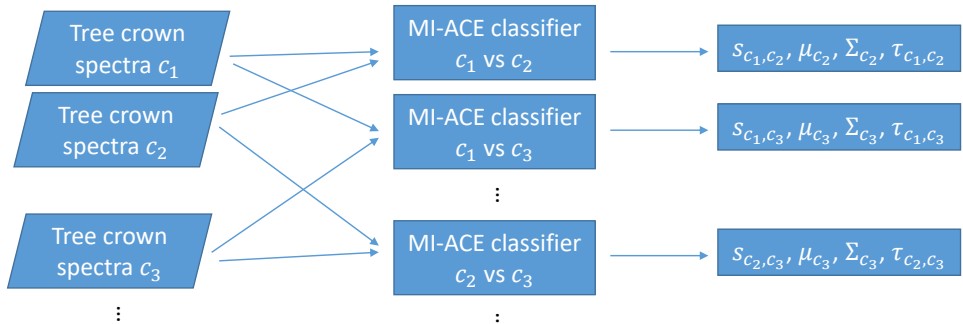

**Figure 1** Training using one-vs-one MI-ACE algorithm.

As outlined by *Zare, Jiao & Glenn (2018)*, the MI-ACE algorithm optimizes Eq. (3) using an alternating optimization strategy. After optimization, an estimate for a discriminative target signature (that is used to distinguish between two classes and perform pixel level classification using the $D_{ACE}$ detector) is obtained. This iterative approach does not require any parameter tuning. The MI-ACE code is available and published on our GitHub site (*Zare, Glenn & Gader, 2018*).

## Proposed one-vs-one MI-ACE

The original MI-ACE algorithm was designed for target detection. Target detection can also be viewed as a two-class classification problem with one class being target and the other class being non-target or background (often with heavily imbalanced class sizes). In this work, we investigate whether MI-ACE can be extended to multi-class classification problems and applied to tree crown species classification using a one-vs-one classification strategy. Two MI-ACE classifiers are trained for every pair of two classes in the multi-class classification problem. Two classifiers are trained so that each class can be considered as the target class once in this pair. An MI-ACE classifier consists of a trained discriminative target signature (estimated using the MI-ACE approach outlined in the previous section), a background mean and covariance computed using the training samples from the non-target class, and a threshold value used to assign a target or non-target label to individual data points given their ACE detection confidence computed using the estimated target signature, background mean and background covariance values, as is shown in Fig. 1.

During testing, each trained MI-ACE classifier is applied to an input test point. Since each classifier yields a classification result, the final classification for a testing bag is obtained by aggregating all of the individual results. Specifically, a test bag is assigned the class label associated with the class that had the largest number of votes associated with each class. The votes are tallied by, first, averaging the confidence values estimated from each of the individual classifiers applied to each test point in the bag and, then, thresholding these average confidence values to obtain a binary target vs. non-target label the test bag. Then, the class label with the largest number of votes from the binary classification results is assigned as the final class labe, as is shown in Fig. 2.
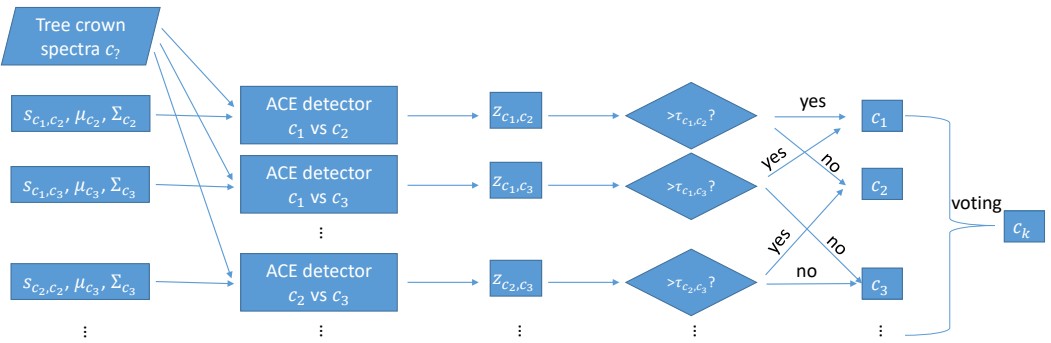

**Figure 2** Testing using one-vs-one MI-ACE algorithm.

Pseudocode for the proposed method is shown below. In the pseudocode, let $\mathbf{X}$ and $\mathbf{Y}$ be the set of all training and testing bags, respectively, with $\mathbf{X}_L$ being the set of all bags assigned label $L$, $\mathbf{Y}_m$ being the $m$th testing bag, and $\mathbf{Y}_{m,n}$ being the $n$th data point in the $m$th testing bag. Let $\mathbf{L}$ and $\mathbf{R}$ be the corresponding bag level labels for the training and testing bags, respectively. $C$ denotes the number of classes. $M$ denotes the number of testing bags. The variables $\mathbf{s}_{i,j}$ and $\tau_{i,j}$ represent the estimated target signature and classification threshold for target class i and background class j, respectively, and $z_{m,i,j}$ denotes the confidence value estimated by ACE detector given the estimated $\mathbf{s}_{i,j}$ and $\tau_{i,j}$ values for the $m$th test bag. The threshold value is set by determining the threshold that minimizes classification error on the training data. This approach does not have any parameters to tune as all parameters are estimated from the training data.

The threshold value, $\tau_{i,j}$, is estimated from the training data by identifying the threshold that minmizes classification error on the training data. After training a set of one-vs-one classifiers as outlined in the psuedo-code, the estimated target signature $\mathbf{s}_{i,j}$ is used to apply the ACE detector to the training data to compute their associated confidence scores. Then, $\tau_{i,j}$ is varied between the largest and smallest computed confidence values by first sorting the confidence values and setting the threshold to be half-way between every pair of consecutive sorted confidence values. At each of these locations, the classification accuracy of the training data is computed. The threshold is set at the value with the maximum classification accuracy on the training data.

## Data description

The training data released by *Marconi et al. (2018)* contains the hyperspectral signatures from 305 tree crowns collected over the Ordway-Swisher Biological Station (OSBS) by the National Ecological Observatory Network (NEON). The data from NEON included the following data products: (1) field data from years 2015-2017: Woody plant vegetation structure (NEON.DP1.10098); (2) hyperspectral data from year 2014: Spectrometer orthorectified surface directional reflectance - flightline (NEON.DP1.30008); (3) LiDAR point cloud data from year 2014: Ecosystem structure (NEON.DP3.30015); and RGB data from year 2014: High-resolution orthorectified camera imagery (NEON.DP1.30010). In this study, only hyperspectral data are used.

---

**Procedure 1** One-vs-one MI-ACE classification

1: *Train two MI-ACE classifiers for each pair of class labels:*

**Input: X, Y, L**

2: **for** Every pair of classes $c_1 = 1 : C$ and $c_2 = 1 : C$ where $c_2 \neq c_1$ **do**

3:      Train MI-ACE: $(\mathbf{s}_{c_1,c_2}, \tau_{c_1,c_2}, \mu_{c_2}, \Sigma_{c_2}) = \text{MI-ACE}(X_{L=c1}, X_{L=c2})$

4:      Train MI-ACE: $(\mathbf{s}_{c_2,c_1}, \tau_{c_2,c_1}, \mu_{c_1}, \Sigma_{c_1}) = \text{MI-ACE}(X_{L=c2}, X_{L=c1})$

5: **end for**

6: *Test using a one-vs-one voting scheme:*

7: **for** Every test bag $m = 1 : M$ **do**

8:      **for** Every pair of classes $c_1 = 1 : C$ and $c_2 = 1 : C$ where $c_2 \neq c_1$ **do**

9:          **for** Every data point, $n = 1 : N_m$ in bag $m$ **do**

10:              Apply the $c_1$ vs. $c_2$ classifier: $z_{m,n,c_1,c_2} = \text{ACE}(\mathbf{Y}_{m,n}, \mathbf{s}_{c_1,c_2})$

11:          **end for**

12:          Average the confidence scores over all points in the bag: $z_{m,c_1,c_2} = \frac{1}{N} \sum_{n=1}^{N_m} z_{m,n,c_1,c_2}$

13:          **if** $z_{m,c_1,c_2} > \tau_{c_1,c_2}$ **then**

14:              $R_{m,c_1,c_2}$ gets one vote for class $c_1$

15:          **else**

16:              $R_{m,c_1,c_2}$ gets one vote for class $c_2$

17:          **end if**

18:      **end for**

19:      $R_m$ is assigned to the class with the largest number of votes.

20: **end for**

**Output: R**

---

Figure 3 shows a region in OSBS containing various tree species from one NEON flight path. The left image in Fig. 3 is the high resolution RGB image collected from NEON AOP with a 0.25 m resolution. The middle image in Fig. 3 shows the maximum height of the canopy top from LiDAR point cloud with a 1 m resolution. The right image is the RGB bands (wavelength 668 nm, 548 nm, 473 nm for red, green and blue) extracted from hyperspectral image from NEON AOP using NEON imaging spectrometer with a 1 m spatial resolution and 5 nm spectral resolution. The training tree crowns in this challenge were extracted by the organizer from a series of regions similar to Fig. 3 from several flight paths. However, for this tree crown classification challenge, only the individual spectral signature of each pixel in each tree crown are stored and provided. Each tree crown usually contains dozens of pixels. Thus, no spatial information is given nor can any image processing approach be applied. Each training tree crown is associated with a genus and species label. The tree crown is a manually selected image patch which may contain, not only the labeled tree pixels, but also some noise pixels or pixels corresponding to sand, neighboring trees or grass growing underneath the labeled tree. Therefore, the given crown level label is imprecise for the individual pixels of which it is composed. The signatures provided contain 426 spectral bands ranging from 383 nm to 2,512 nm. Water absorption wavelengths, of which reflectance are set to be 1.5 as shown in Fig. 4,
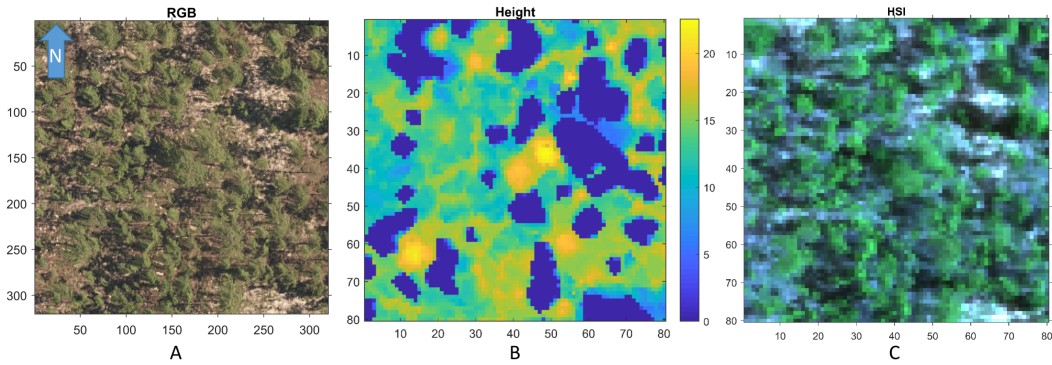

**Figure 3** An example RGB (A), LiDAR (B), and (the RGB image generated from the corresponding) Hyperspectral image (C) of a region in OSBS. Both axes are in meters.

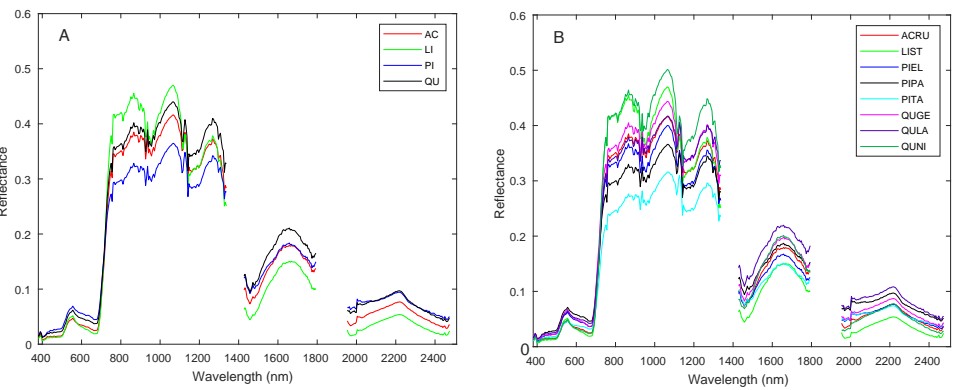

**Figure 4** Average spectral signature of (A) all genera and (B) all species, colored by genera.

correspond to 1,345 nm to 1,430 nm, 1,800 nm to 1,0956 nm and 2,482 nm to 2,512 nm. Each training tree crown is paired with a genus class label and a species class label. The genus consisted of 5 classes which are *Acer* (AC), *Liquidambar* (LI), *Pinus* (PI), *Quercus* (QU) and OTHERS (OT). OTHERS represent the tree crowns that cannot be classified to any one of the four known genera. Each genus has a different number of associated species. AC and LI contains only one species, which are *Acer rubrum* (ACRU) and *Liquidambar styraciflua* (LIST), respectively. PI and QU contain more than three species individually, which are *Pinus elliottii* (PIEL), *Pinus palustris* (PIPA), *Pinus taeda* (PITA) and OTHERS (for PI), *Quercus geminata* (QUGE), *Quercus laevis* (QULA), *Quercus nigra* (QUNI) and OTHERS (for QU). The number of tree polygons for each species are shown in Table 1. There are both prevalent species, such as PIPA and QULA, and rare species, such as ACRU and LIST in the dataset, which make the competition more challenging. In the current implementation of this work, OTHERS in both the genus and species level are not used for

**Table 1  The number of training tree crowns for each species**

| Species | ACRU | LIST | PIEL | PIPA | PITA | QUGE | QULA | QUNI | OTHERS |
|---|---|---|---|---|---|---|---|---|---|
| Number | 6 | 4 | 5 | 197 | 14 | 12 | 54 | 5 | 8 |

training as the proposed approach did not have a mechanism to identify points that did not belong to any of the labeled training classes.

The testing data are also NEON tree crown hyperspectral data in the same format. There were 126 testing tree crowns. The test labels were not provided by the competition organizers.

## Data preprocessing and MI-ACE training

Prior to application of the MI-ACE algorithm, the water bands were removed to minimize the influence of noise and reduce dimensionality of the dataset. There were in total 55 water bands that are removed, correspond to wavelength of 1,345 nm to 1,430 nm, 1,800 nm to 1,956 nm and 2,482 nm to 2,512 nm. Then, since in our current implementation data points labeled as OTHERS genus or species are not addressed, the signatures that were labeled as OTHERS are removed from the training set.

After removal of the water bands and the OTHERS data points, the target signatures and classification threshold values are trained using the one-vs-one MI-ACE classification technique. Training was conducted at two levels, the genus and the species levels. During the training phase, each training tree crown was considered a bag for MI-ACE, thus the training label (genus or species level) was the bag label. A one-vs-one MI-ACE was used in which a set of MI-ACE target signatures representing the difference between every two genera or species were estimated. For instance, a target signature was trained to distinguish between the genus PI and the genus QU where tree crown labeled as PI was labeled target (or '1') and QU was labeled as non-target (or '0'). For this competition only one MI-ACE classifier was trained for each pair of classes (as opposed to two for each pair) because results were similar between the two approaches. Similarly, a set of target signatures were estimated between every two species (if there were at least two species) that belonged to the same genus. For example, a target signature was estimated using training data to distinguish between species PIEL and PITA where the tree crowns labeled as PIEL were labeled as target (or '1') and PITA was labeled as non-target (or '0').

## Testing using ACE detector and voting

Testing was also conducted in two stages where a test tree crown was first classified at the genus level and, then, further classified at the species level. An ACE detector was used to estimate the confidence value indicating how similar a test signature is to a trained target signature. Classification of test tree crowns consisted of the following steps. First, the confidence values of each instance signature inside the a test tree crown were computed using each of the six genus-level trained MI-ACE classifiers. Second, the confidence value for each testing crown was estimated by taking the average value over all of the instance-level confidence values. These average confidence values were then thresholded using the trained threshold values to obtain a binary classification result. The classification thresholds were

determined during training to minimize the number of misclassified tree crowns in the training data. The final classification of a tree crown is the class label with the highest number of corresponding binary classifications. After the genus level classification, a test tree crown can be classified at species level using the same approach among the species associated with the genus to which the tree crown assigned.

There are three types of experimental validation methods used in this study: test-on-train, test-on-test and cross validation. Suppose the training dataset and testing dataset are $X$, $Y$, respectively. Test-on-train experiments represent training the model using $X$ and testing also using $X$. Test-on-test experiments represent training the model using $X$ and testing using $Y$. Cross validation experiments represent training the model using $d_1$ and validate using $d_2$, then train using $d_2$ and validate using $d_1$, where $X = \{d_1, d_2\}$, $d_1$ and $d_2$ are the subsets of the training data $X$. Test-on-train and cross-validation experiments are included since groundtruth labels for the test data have not been released by competition organizers.

## Evaluation metrics

The metrics used by competition organizers to evaluate performance are rank-1 accuracy, cross entropy, accuracy and specificity score, F1 score, precision and recall. The rank-1 accuracy is the fraction of crowns whose ground truth class is assigned the highest probability by the proposed algorithm. Intuitively, rank-1 accuracy only consider the percentage of crowns that are correctly predicted (assigned with highest probability). The rank-1 accuracy, $r$, is defined as:

$$r = \frac{1}{N} \sum_{i=1}^{N} \delta\left(\underset{k}{\arg\max}(p_{i,k}), g_i\right) \tag{6}$$

where $g_i$ is the ground truth class of crown $i$ and $p_{i,k}$ is the probability assigned that crown $i$ belongs to class $k$. The $\delta(x,y)$ is an indicator function that takes value 1 if $x = y$ and 0 otherwise. The cross entropy, $c$, is defined as:

$$c = -\frac{\sum_{i,k} \delta(k, g_i) \ln p_{i,k}}{N} \tag{7}$$

where $g_i$ is the ground truth class of crown $i$ and $p_{i,k}$ is the probability assigned that crown $i$ belongs to class $k$. The $\delta(x,y)$ is an indicator function that takes value 1 if $x = y$ and 0 otherwise. The accuracy for $k$th class, $a_k$, is defined as:

$$a_k = \frac{TP_k + TN_k}{TP_k + TN_k + FP_k + FN_k} \tag{8}$$

where $TP_k$, $TN_k$, $FP_k$ and $FN_k$ represents the true positive, true negative, false positive and false negative for class $k$, respectively. The specificity for $k$th class, $s_k$, is defined as:

$$s_k = \frac{TN_k}{TN_k + FP_k} \tag{9}$$

where $TN_k$ and $FP_k$ represents the true negative and false positive for class $k$, respectively. The precision for $k$th class, $p_k$, is defined as:

$$p_k = \frac{TP_k}{TP_k + FP_k} \tag{10}$$
where $TP_k$ and $FP_k$ represents the true positive and false positive for class $k$, respectively. The recall for $k$th class, $e_k$, is defined as:

$$e_k = \frac{TP_k}{TP_k + FN_k} \tag{11}$$

where $TP_k$ and $FN_k$ represents the true positive and false negative for class $k$, respectively. The F-1 score for $k$th class, $f_k$, is defined as:

$$f_k = 2\frac{p_k e_k}{p_k + e_k} \tag{12}$$

where $p_k$ and $e_k$ represents the precision and recall for class $k$, respectively.

## EXPERIMENTAL RESULTS

The extended MI-ACE method, one-vs-one MI-ACE, was applied to and entered into the tree crown classification challenge organized and described by *Marconi et al. (2018)*.

### Genus level classification

Classification result when testing on training samples are shown via confusion matrix in Tables 2 and 3 showing results where one classifier is trained for each pair of species classes and two classifiers are trained for each pair of species classes, respectively. The overall classification accuracy on the training dataset is 97.31% for both the one classifier and two classifiers cases. The pixel confidence distributions and the aggregated crown confidence distributions for each classifier are shown in Figs. 5 and 6 respectively. In Figs. 5 and 6, each subfigure represents the confidence distribution of tree crowns from one ground truth genus type estimated by one of the six classifiers . The associated threshold value for each classifier is shown as a vertical blue line in each subfigure. For instance, the subfigure A in Fig. 5 shows the averaged ACE confidence value distribution of AC tree crowns detected using AC-vs-LI classifier and the subfigure A in Fig. 6 shows the corresponding pixel confidence value distributions. A high classification accuracy result for the AC-vs-LI classifier will have all AC confidence values to right of the threshold value and all LI confidence values (shown in the subfigure B) to the left of the threshold value. As can be seen, the AC-vs-LI classifier accurately distinguishes between these two classes on the aggregated crown-level scale. However, when considering the same plots in Fig. 6 for the pixel level confidences, we can see that there are many AC pixels to the left of the threshold causing significant overlap with the LI pixels confidences. This indicates that the aggregation procedure helps to improve results.

Since there were six classifiers trained and four classes in this data set, there are only a small set of possible voting cases. These cases are: a. (3,1,1,1) votes for each class; b. (3,2,1,0) votes for each class; c. (2,2,1,1) votes for each class and d. (2,2,2,0) votes for each class. For cases a. and b., there is a single class with the largest amount of votes, thus, labeling of the crown is straightforward. However, for voting cases c. and d., there are ties among 2 or 3 candidate classes. In our current implementation, we randomly assign the label of one of the tied classes. We found that cases c. and d. are rare in our training and testing results. In the testing on training data results, votes for all of the tree crowns fell into either case a. or

**Table 2   The classification confusion matrix on all training data (except for OTHERS) in genus level (one classifier per pair).**

| True/Predict | AC | LI | PI | QU |
|---|---|---|---|---|
| AC | 6 | 0 | 0 | 0 |
| LI | 0 | 4 | 0 | 0 |
| PI | 0 | 0 | 212 | 4 |
| QU | 0 | 0 | 4 | 67 |

**Table 3   The classification confusion matrix on all training data (except for OTHERS) in genus level (two classifiers per pair).**

| True/Predict | AC | LI | PI | QU |
|---|---|---|---|---|
| AC | 6 | 0 | 0 | 0 |
| LI | 0 | 4 | 0 | 0 |
| PI | 1 | 0 | 214 | 1 |
| QU | 1 | 0 | 5 | 65 |

b. When applying the trained approach to the testing dataset provided by the competition, there is only one tree crown in which there was a tie (and resulted in voting case is d). For this tree crown, we found that it has two votes for AC, two votes for LI and two votes for PI. Our implementation in test randomly assigned this tree crown to the AC class. Since the ground truth labels of testing data are not released by organizer, the true genus class of this testing tree crown is unknown. However, we found that in our classification results on testing data, there are three tree crowns that were predicted to be in class AC by our method. Two of these crowns were correctly classified into class AC and the other false positive tree crown actually belonged to the OTHER class. Thus, it is likely that this tree crown may be the tree crown with the tied result.

Cross validation studies on the training data were also conducted. There are a limited number of training tree crowns for the AC and LI classes (only six AC and four LI tree crowns). Due to this reason, cross validation experiments were not conducted for AC or LI. In the training phase, the PI training (pixel-level) samples and QU (pixel-level) training samples are considered target and background, respectively. The learned target signature is shown in Fig. 7A, which characterizes the spectral difference between PI and QU shown in Fig. 7B. In the testing phase, each pixel signature of PI and QU are compared with estimated target signature using the ACE detector resulting in a confidence value shown in Fig. 8. Since most confidence values of PI pixels are larger than QU pixels, a threshold value (of 0.05) can be selected such that the misclassified error is minimized.

Two-fold cross validation was applied to the PI and QU samples. The PI and QU classes were randomly split into two datasets (50% of PI and QU tree crowns are selected as $d_1$ and the rest as $d_2$). We train on $d_1$ and validate on $d_2$, followed by training on $d_2$ and validating on $d_1$. After training the PI-vs-QU target signature and corresponding threshold, the histogram of the average confidence values on the validation set is shown in Fig. 9. In this figure, the PI and QU tree crowns are colored by their ground truth classes, i.e., red for

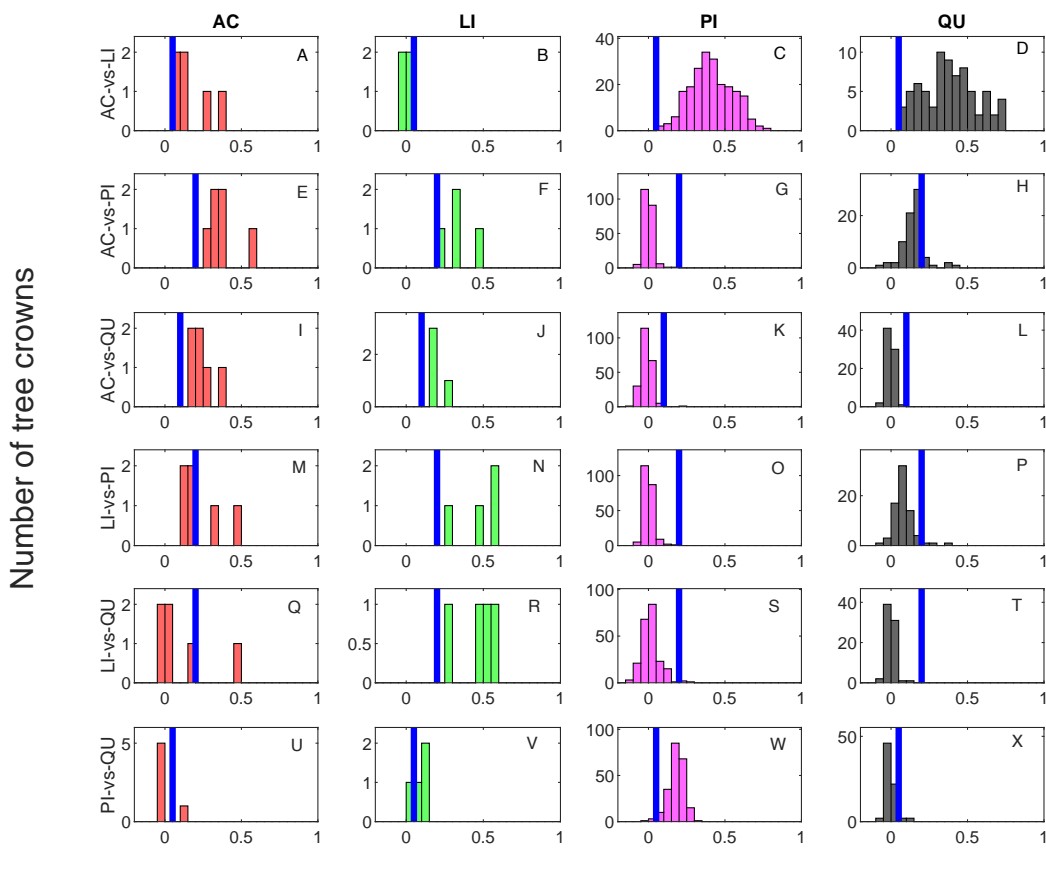

Number of tree crowns

Confidence

**Figure 5  Confidence distributions of crown levels in training set.** Blue vertical lines are the corresponding estimated threshold values. (A) AC, (B) LI, (C) PI and (D) QU tree crowns detected using AC-vs-LI classifier; (E) AC, (F) LI, (G) PI and (H) QU tree crowns detected using AC-vs-PI classifier; (I) AC, (J) LI, (K) PI and (L) QU tree crowns detected using AC-vs-QU classifier; (M) AC, (N) LI, (O) PI and (P) QU tree crowns detected using LI-vs-PI classifier; (Q) AC, (R) LI, (S) PI and (T) QU tree crowns detected using LI-vs-QU classifier; (U) AC, (V) LI, (W) PI and (X) QU tree crowns detected using PI-vs-QU classifier.

PI and blue for QU. The threshold value estimated from training can be directly applied to the validation set for classification of the validation training crowns.

The cross validation experiment was repeated ten times and the mean confusion matrix is shown in Tables 4 and 5 where one classifier is trained for each pair of species classes and two classifiers are trained for each pair of species classes, respectively. The average classification accuracy on the PI and QU given two-fold cross validation dataset was 95.8% and 96.78% for the one classifier and two classifiers cases, respectively, which is similar to the test-on-train accuracy indicating robust results. From the genus level classification, training two classifiers for each pair of classes can achieve classification accuracy slightly higher or equal to the training one classifier for each pair of classes.

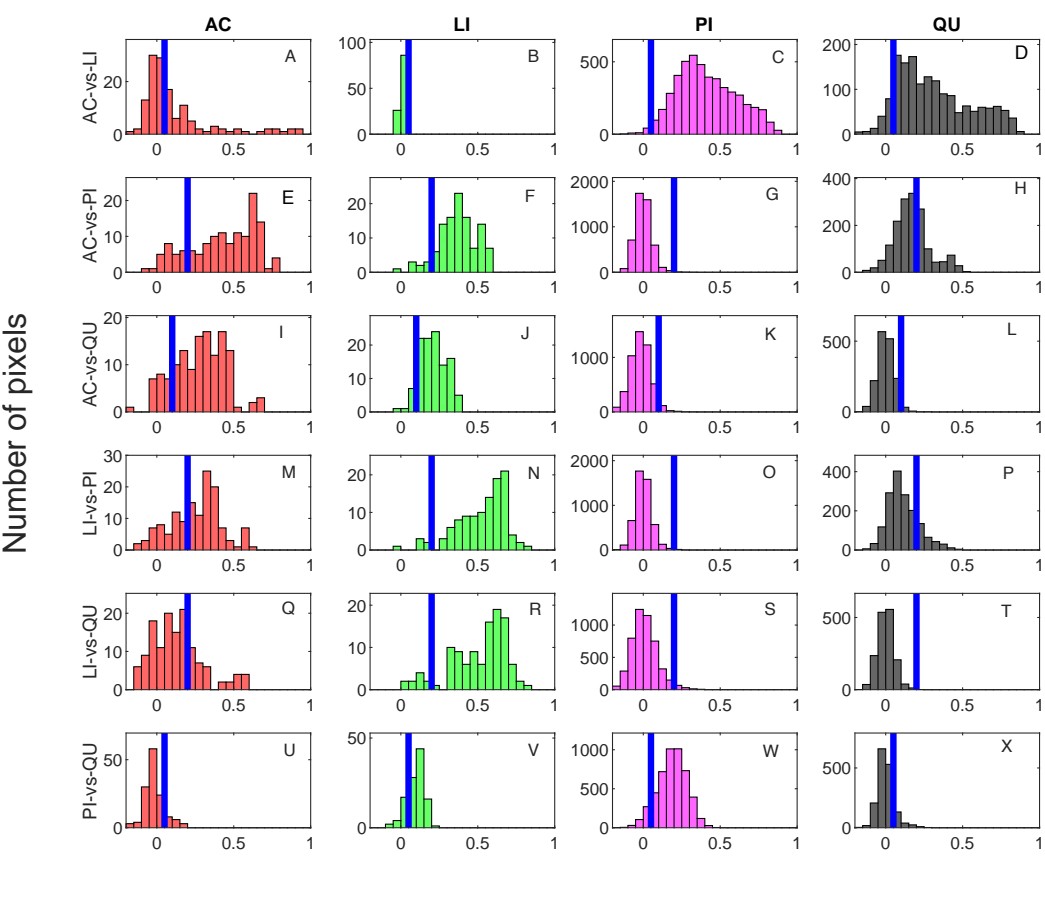

**Figure 6** **Confidence distributions of pixel levels in training set.** Blue vertical lines are the corresponding estimated threshold values. (A) AC, (B) LI, (C) PI and (D) QU tree crowns detected using AC-vs-LI classifier; (E) AC, (F) LI, (G) PI and (H) QU tree crowns detected using AC-vs-PI classifier; (I) AC, (J) LI, (K) PI and (L) QU tree crowns detected using AC-vs-QU classifier; (M) AC, (N) LI, (O) PI and (P) QU tree crowns detected using LI-vs-PI classifier; (Q) AC, (R) LI, (S) PI and (T) QU tree crowns detected using LI-vs-QU classifier; (U) AC, (V) LI, (W) PI and (X) QU tree crowns detected using PI-vs-QU classifier.

## Species level classification

After the genus level classification, the tree crowns were further classified into species. If a tree is classified as AC or LI, it is classified also as ACRU or LIST automatically. If a tree is classified as PI or QU, the one-vs-one MI-ACE method is used to classify it into one of the corresponding species. The confusion matrices for species level classification (testing on training data) are shown in Table 6. The classification (rank-1) accuracy is 95.62% on the training dataset in species level with a cross entropy value of 0.2649.

The confusion matrices for species level classification for testing data are shown in Fig. 10 as provided by the competition organizers. The classification (rank-1) accuracy is 86.40% and cross entropy is 0.9395 on the testing dataset. However, this accuracy includes data points labeled as OTHERS in the testing dataset which, using our approach, were all

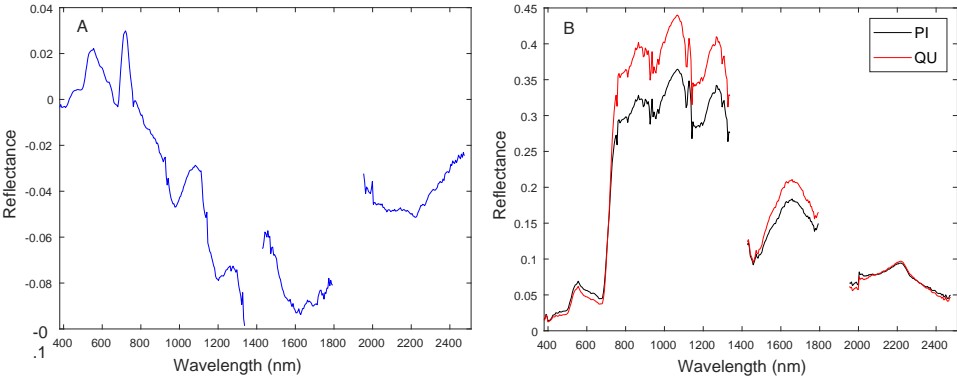

**Figure 7** **Comparison between estimated target signature (A) and average class signatures (B).** As can be seen, the target signature tends to be positive in the wavelengths where the target class has a larger response than the background class and negative where the target class has a smaller response. Thus, the target signature gives insight into discriminative features for the detection problem.

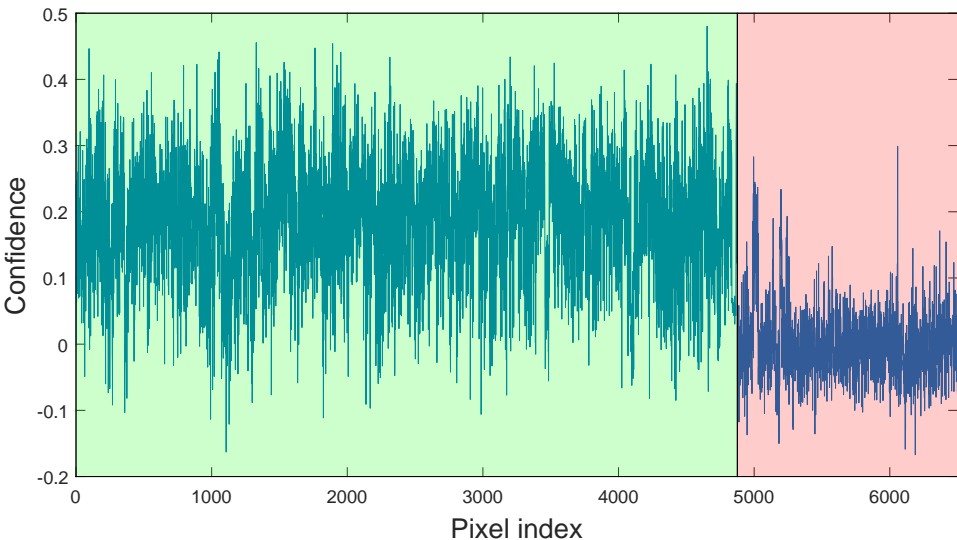

**Figure 8** **ACE detection statistic on PI & QU pixels (green (first set): PI pixels; pink (second set): QU pixels).**

misclassified to one of the four genus types since we did not implement a mechanism to distinguish outliers in this approach. If the OTHERS tree crowns (three tree crowns) are excluded, the classification accuracy would come to 88.52% and cross entropy would be 0.7918 on the testing dataset.

The species level classification results are further evaluated using several metrics on the testing data by the organizer, including per-class accuracy, specificity, precision, recall and F1 score. For comparison, we also evaluated the classification performance using the same metrics on the training data. The accuracy and specificity score, F1 score, precision, recall for both training and testing dataset are shown in Figs. 11–14, respectively. As can be seen,

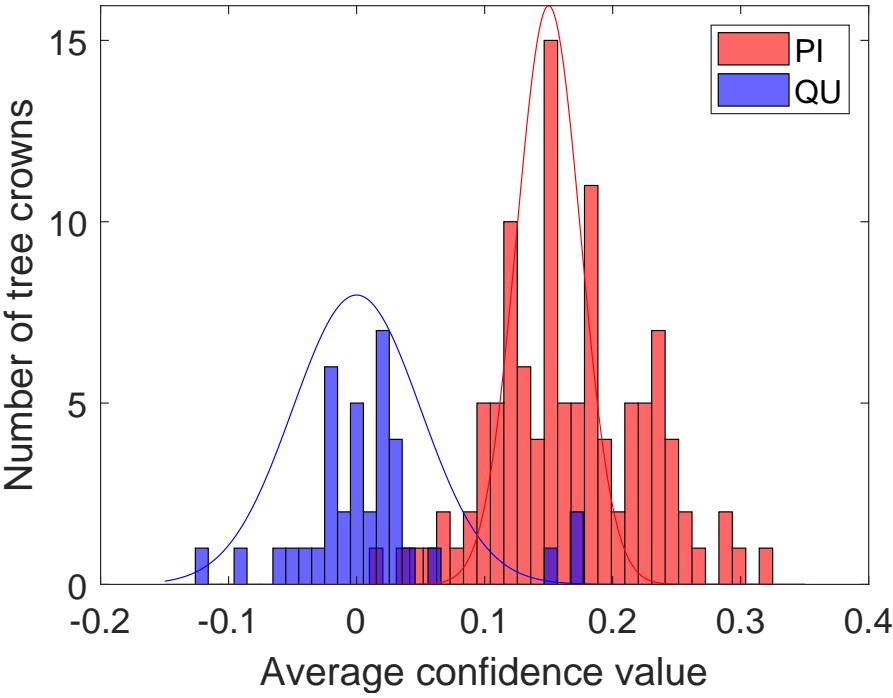

**Figure 9** Histogram of average confidence values on validation set (red: PI; blue: QU).

**Table 4  The mean classification confusion matrix on all PI and QU training data via cross validation (one classifier per pair).**

| True/Predict | PI | QU |
|---|---|---|
| PI | 105.8 | 2.2 |
| QU | 3.8 | 31.2 |

**Table 5  The mean classification confusion matrix on all PI and QU training data via cross validation (two classifiers per pair).**

| True/Predict | PI | QU |
|---|---|---|
| PI | 106.8 | 1.2 |
| QU | 3.4 | 31.6 |

accuracy and specificity results between training and testing data are similar whereas the F1, precision and recall curves highlight that challenging classes in the testing data were PIEL, PITA and QUGE.

## Discussions

In the current implementation, only crisp binary classification results are estimated. However, competition organizers evaluated results using the cross entropy evaluation metric assuming probabilities of belonging to each class are estimated, following Eq. (7). Class probabilities given the one-vs-one scheme can be estimated in the future using

**Table 6** The classification confusion matrix on all training data (except for OTHERS) in species level.

| True/Predict | ACRU | LIST | PIEL | PIPA | PITA | QUGE | QULA | QUNI |
|---|---|---|---|---|---|---|---|---|
| ACRU | **6** | 0 | 0 | 0 | 0 | 0 | 0 | 0 |
| LIST | 0 | **4** | 0 | 0 | 0 | 0 | 0 | 0 |
| PIEL | 0 | 0 | **5** | 0 | 0 | 0 | 0 | 0 |
| PIPA | 0 | 0 | 3 | **188** | 2 | 2 | 2 | 0 |
| PITA | 0 | 0 | 0 | 0 | **14** | 0 | 0 | 0 |
| QUGE | 0 | 0 | 0 | 2 | 0 | **10** | 0 | 0 |
| QULA | 0 | 0 | 0 | 2 | 0 | 0 | **53** | 0 |
| QUNI | 0 | 0 | 1 | 0 | 0 | 0 | 0 | **4** |

**Notes.**
Bold styling emphasizes the number of samples that are correctly predicted.

| Species ID | ACRU | LIST | OTHER | PIEL | PIPA | PITA | QUGE | QULA | QUNI |
|---|---|---|---|---|---|---|---|---|---|
| ACRU | **2.00** | 0.00 | 0.00 | 0.00 | 0.00 | 0.00 | 0.00 | 0.00 | 0.00 |
| LIST | 0.00 | **1.00** | 0.00 | 0.00 | 0.00 | 0.00 | 0.00 | 0.00 | 0.00 |
| OTHER | 1.00 | 1.00 | **0.00** | 1.00 | 0.00 | 0.00 | 0.00 | 0.00 | 0.00 |
| PIEL | 0.00 | 0.00 | 0.00 | **1.00** | 0.00 | 0.00 | 1.00 | 0.00 | 0.00 |
| PIPA | 0.00 | 0.00 | 0.00 | 1.00 | **81.00** | 0.00 | 1.00 | 0.00 | 0.00 |
| PITA | 0.00 | 0.00 | 0.00 | 1.00 | 2.00 | **2.00** | 0.00 | 1.00 | 0.00 |
| QUGE | 0.00 | 1.00 | 0.00 | 0.00 | 0.00 | 0.00 | **3.00** | 0.00 | 0.00 |
| QULA | 0.00 | 0.00 | 0.00 | 0.00 | 1.00 | 0.00 | 4.00 | **17.00** | 1.00 |
| QUNI | 0.00 | 0.00 | 0.00 | 0.00 | 0.00 | 0.00 | 0.00 | 0.00 | **1.00** |

**Figure 10** The classification confusion matrix on all testing data (except for OTHERS) in species level (provided by competition).

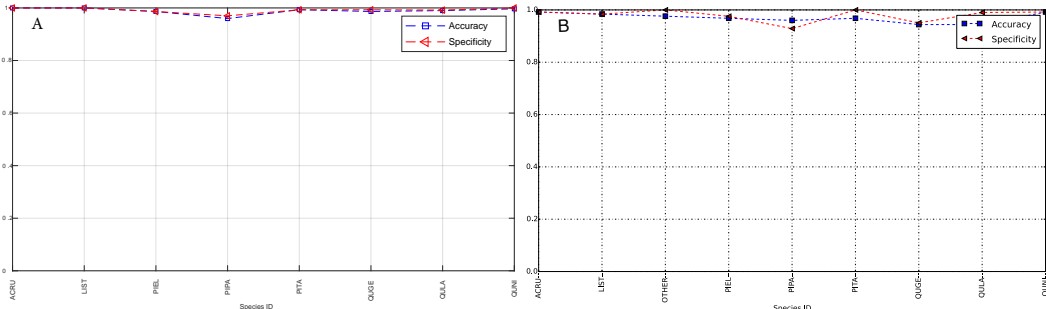

**Figure 11** Accuracy and Specificity Scores (Per-Class) for training data (A) and testing data (B—provided by competition).

approaches such as those proposed by *Wu, Lin & Weng (2004)*. Furthermore, even if individual probabilities per data are not computed, an overall uncertainty value can be estimated from the training data. In other words, as opposed to assigning 0-1 probabilities for the crisp class labels. In our implementation data points were assigned to the estimated

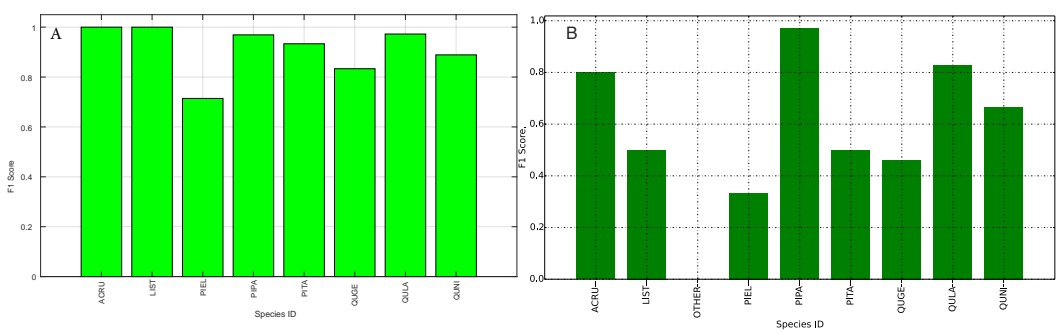

**Figure 12 F1 Scores (Per-Class) for training data (A) and testing data (B—provided by competition).**

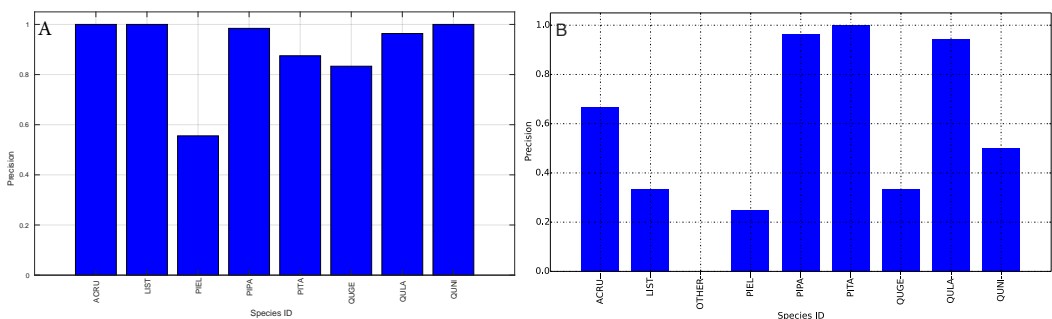

**Figure 13 Precision (Per-Class) for training data (A) and testing data (B—provided by competition).**

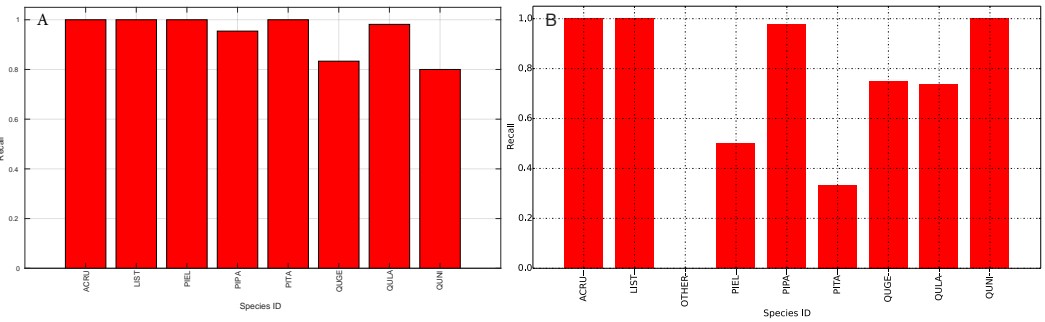

**Figure 14 Recall (Per-Class) for training data (A) and testing data (B—provided by competition).**

class label with probability 1 and all others with probability 0. Instead, we could pre-compute an optimal epsilon value, $\epsilon^*$, to add to the '0' probabilities and subtract from the '1' probabilities to ensure values sum to one across classes to minimize cross entropy on the training data. For instance, we found that when $\epsilon^* = 0.017$, the cross entropy for our results comes to 0.68, which is a smaller (i.e., better) than the cross entropy of 0.94 obtained

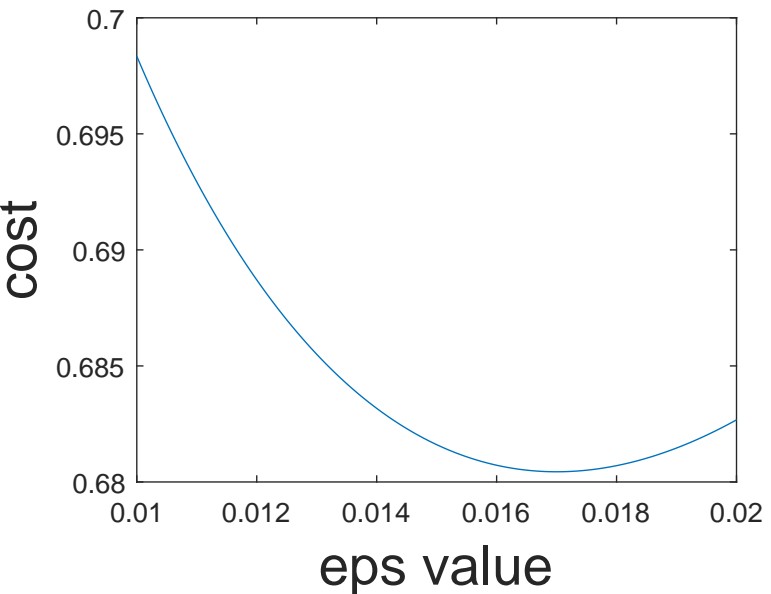

**Figure 15** Cross entropy vs optimum epsilon value.

using crisp labels and calculated by the competition organizers. The relationship between the cross entropy and epsilon value for the training data provided shown in Fig. 15.

In addition, in the current proposed framework, each one-vs-one classifier is equally weighted in final voting. However, the classification accuracies and the applicability of each classifier varies. For instance, if a tree crown is in class PI, the PI-vs-QU classifier should be more heavily weighted than the AC-vs-LI classifier. Investigation into whether this could be determined by considering the average confidence values estimated from the individual ACE detectors is needed. Furthermore, since some of the classes are more spectrally distinct, some one-vs-one classifiers have better prediction performances. One possible solution is to weight the classifiers based on the difference between the average confidence values of target and background classes for training data. Another possible solution is to weight based on the difference between the confidence value of testing point and threshold value of the classifier. In some scenarios, these two solutions might be equivalent. Finally, data fusion is also a promising approach for boosting the classification performance. For example, height information from Lidar data could be also incorporated into the training phase since different species generally have different average heights. In the current implementation, only hyperspectral information was leveraged.

## CONCLUSIONS

A one-vs-one version of MI-ACE is proposed in the work to address the hyperspectral tree crown classification problem. The proposed method achieved a 86.4% overall classification accuracy on a blind testing dataset. The 95.62% and 86.4% species level classification accuracy on training and testing data, respectively, show that the multiple instance learning based algorithm, one-vs-one MI-ACE is capable of learning the discriminative spectrum

difference among different species classes and allowing for the weakly supervised, imprecise, crown-level labels. Certainly, there are many improvements can be investigated in the future such as mechanisms to identify outliers and label them as members of the OTHERS class and estimate a likelihood of belonging to each class (as opposed to binary classification labels).

### Funding

This material is based upon work supported by the National Science Foundation under Grant IIS-1723891-CAREER: Supervised Learning for Incomplete and Uncertain Data. The National Ecological Observatory Network is a program sponsored by the National Science Foundation and operated under cooperative agreement by Battelle Memorial Institute. This material is based in part upon work supported by the National Science Foundation through the NEON Program. The ECODSE competition was supported, in part, by a research grant from NIST IAD Data Science Research Program to D Z Wang, E P White, and S Bohlman, by the Gordon and Betty Moore Foundation's Data-Driven Discovery Initiative through grant GBMF4563 to E P White, and by an NSF Dimension of Biodiversity program grant (DEB-1442280) to S Bohlman. There was no additional external funding received for this study. The funders had no role in study design, data collection and analysis, decision to publish, or preparation of the manuscript.

### Grant Disclosures

The following grant information was disclosed by the authors:
National Science Foundation: IIS-1723891-CAREER.
The National Ecological Observatory Network.
Battelle Memorial Institute.
National Science Foundation.
NIST IAD Data Science Research Program.
Gordon and Betty Moore Foundation's Data-Driven Discovery Initiative: GBMF4563.
NSF Dimension of Biodiversity: DEB-1442280.

### Competing Interests

The authors declare there are no competing interests.

### Author Contributions

- Sheng Zou conceived and designed the experiments, performed the experiments, analyzed the data, prepared figures and/or tables, authored or reviewed drafts of the paper.
- Paul Gader conceived and designed the experiments, analyzed the data.
- Alina Zare conceived and designed the experiments, analyzed the data, authored or reviewed drafts of the paper, approved the final draft.

### Data Availability

Hyperspectral Toolkit Code: Available at https://zenodo.org/record/1260272

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
