# Peer review of "Hyperspectral tree crown classification using the multiple instance adaptive cosine estimator"

_PeerJ, doi:10.7717/peerj.6405_

## Round 0.1 · original submission · Major Revisions

I have now received two detailed reviews of this paper. The reviewers suggested a number of revisions and clarifications that will add to the scientific value of this paper. In particular, I recommend that you pay close attention to Reviewer 2 and the request for a bit more context. Also please correct the many grammatical errors noted by the reviewers.

Reviewer 1 ·

Basic reporting

The manuscript is very well written and conceived – with sufficient context given where needed. I like the overall structure of the manuscript in that it does not focus on the NIST competition itself but rather focuses in describing the method or science behind it.

Experimental design

The paper describes a neural network algorithm to classify trees using hyperspectral data. This method supposedly addresses the issue of mislabeled pixels.

The algorithm is described in detail. The core algorithm is also described in a separate work (Zare et al in press at IEEE) and is open source (available at GitHub). In this manuscript, the authors extend the core algorithm to “one-vs-one” – whereby two classifiers are trained for each pair of class label. A pseudocode for the modified algorithm is also provided.

One suggestion to the authors - since “one classifiers for each pair of class label” is applied in the results section, it may make sense to describe here (in the “Proposed one-vs-one MI-ACE” section), “one classifiers for each pair” instead of current “two classifiers for each pair”.

Validity of the findings

This manuscript is significant contribution to the field. The algorithm should be easily adopted to other data or study areas beyond the competition. Unlike other approaches, as the parameters are estimated from the training samples, there is no need for parameter optimization – which is a big advantage of this approach.

Additional comments

Line 38: Correct spelling for “absence”
Line 53: “is” missing in “method that based”
Lines 72: Is it “every pair of classes” or “every pair of two classes”?
Lines 75-77: Please describe in detail how the “threshold value” is computed. Since this is an important step of the classification, an equation or pseudo-code would be helpful.
Lines 98-113: Please describe the hyperspectral data used in more detail – for example, how and when was it collected, what sensor was used, spatial/spectral resolution, etc.
Line 105. Was the individual spectral signature the mean of all pixels within the tree crown? If so, please state.
Line 123. State how many of 426 bands were removed.
Line 162. Change “the first row” to “the top row”
Line 155. The “Data Description” section states that the competition only provided “spectral signatures for each tree crown” and “no spatial information was given”. If so, can the authors describe here how was pixel confidence distributions computed?

Figure 1.
- State which bands were used to generate RGB image.
- Provide a label for the LiDAR image so readers know what which color means which height.
- State what X and Y axis are? If it is longitude, latitude It appears the first is different from the last two.
- State what is HSI, and what bands were used to produce the image.
Figure 2. The labels for Figure 2 (b) show the genus instead of species. It should be ACRU, LIST, etc.
Figure 3. State what the vertical blue lines means in the figure caption.

·

Basic reporting

Clear and unambiguous, professional English used throughout.
- Some sentences need grammatical revision and are highlighted in yellow in the manuscript. The paper should be reviewed for clarity before re-submission.
Literature references, sufficient field background/context provided.
- The Introduction would benefit early on from additional background on the NEON challenge and how and why it came about. This can help tie the work to larger explanation of how the research is relevant and meaningful and fills and identified knowledge gap (see next section, Experimental Design below)
- In addition, some background on the classification algorithm and how it’s design might be better than other algorithms for classifying species. Is the algorithm similar in principal to any other types of machine learning algorithms?
- There are not a lot of references included in the study – some additional background that is narrow in focus and contains references to previous studies would be helpful.
Professional article structure, figs, tables. Raw data shared.
- Abstract: could be expanded from 114 words to up to 500 to include concise descriptions of results and conclusions per abstract journal standards. Also consider Keywords that might help your article appear in more search results (e.g., “NEON”, “species classification”)
- Methods: there is not a clear Methods section and the article will need to be carefully reorganized. Suggest renaming “Proposed Approach” to “Methods” and moving subsections under Experimental Results section such as “Data description” to the “Methods” section.
- This section would also benefit from some sort of a processing flowchart. Also, the code that is provided seems to randomly appear without any background.
Self-contained with relevant results to hypotheses.
-The hypothesis needs to be formulated. Suggest beginning with comment on Line 70. The conclusions section should then be updated to address whether or not results support the hypothesis.

Experimental design

Original primary research within Aims and Scope of the journal.
-Yes
Research question well defined, relevant & meaningful. It is stated how research fills an identified knowledge gap.
-The research question could be better defined and presented earlier on. See comment on line 70. Challenges need to be better developed and more clearly defined. For example, the challenges of “imprecidse labels”, “parameter tuning” and “additional steps for outlier removal” in lines 34 to 35 of the introduction could be expanded to track throughout the paper.
Rigorous investigation performed to a high technical & ethical standard.
- Inputs and image preprocessing steps are unclear. For example, it’s difficult based on Figure 1 and the NEON product descriptions to tell whether or not lidar data were used. See specific comments under Data description.
- It is unclear whether or not validation data that is separate from the training data were used to compute overall classification accuracy since ground truth data were not released in the competition. Along these lines, the number of training samples widely varies for each category, while the rule of thumb is about 50 samples per category (Table 1). This is what was provided in the challenge, but should be pointed out. Validation and Cross validation methods should be described in the Methods section.
Methods described with sufficient detail & information to replicate.
-As of now, no. A start would be to define classification inputs, address whether or not lidar and field data were used and how the classification results were validated. See specific comments in the attached *.pdf.

Validity of the findings

- Can the authors speak to how their classification results compared to results of other competitors, even if based on results that the competitors published? The way the experiment is currently designed, there is no way of comparing results to anything else.
- Do results support a hypothesis or warrant expanded testing – for example, compare performance to another classification method, or extend algorithm to similar datasets acquired in other geographic locations?
- Can the authors identify or discuss any of the study limitations?

Additional comments

Any paper that introduces a new hyperspectral image classification algorithm that is not dependent on subjective thresholding is exciting.It would be great to have some context on why MI-ACE was developed and what classification weaknesses it is designed to address. This may be in the text of other cited work, but worth pulling out.

---

## Round 0.2 · accepted · Accept

Thank you for a well-constructed revision to this paper. I believe it is a valuable contribution to the collection of papers from the NEON-NIST Data Challenge.

·

Basic reporting

Basic reporting criteria satisfied after revisions were made to the article per reviewer comments.

Experimental design

Experimental design criteria satisfied after revisions were made to the article per reviewer comments.

Validity of the findings

Validity of the findings criteria satisfied after revisions were made to the article per reviewer comments.

Additional comments

The article meets PeerJ criteria after revisions in response to reviewer comments.